# Isolation and Investigation of Potential Non-*Saccharomyces* Yeasts to Improve the Volatile Terpene Compounds in Korean Muscat Bailey A Wine

**DOI:** 10.3390/microorganisms8101552

**Published:** 2020-10-08

**Authors:** Sae-Byuk Lee, Heui-Dong Park

**Affiliations:** 1School of Food Science and Biotechnology, Kyungpook National University, 80 Daehakro, Daegu 41566, Korea; lsbyuck@nate.com; 2Institute of Fermentation Biotechnology, Kyungpook National University, 80 Daehakro, Daegu 41566, Korea

**Keywords:** non-*Saccharomyces* yeasts, MBA wine, Fermentation, β-glucosidase, volatile aromatic compound

## Abstract

The Muscat Bailey A (MBA) grape, one of the most prominent grape cultivars in Korea, contains considerable amounts of monoterpene alcohols that have very low odor thresholds and significantly affect the perception of wine aroma. To develop a potential wine starter for Korean MBA wine, nine types of non-*Saccharomyces* yeasts were isolated from various Korean food materials, including *nuruk*, Sémillon grapes, persimmons, and Muscat Bailey A grapes, and their physiological, biochemical, and enzymatic properties were investigated and compared to the conventional wine fermentation strain, *Saccharomyces cerevisiae* W-3. Through API ZYM analysis, *Wickerhamomyces anomalus* JK04, *Hanseniaspora vineae* S7, *Hanseniaspora uvarum* S8, *Candida railenensis* S18, and *Metschnikowia pulcherrima* S36 were revealed to have β-glucosidase activity. Their activities were quantified by culturing in growth medium composed of different carbon sources: 2% glucose, 1% glucose + 1% cellobiose, and 2% cellobiose. *W. anomalus* JK04 and *M. pulcherrima* S36 showed the highest β-glucosidase activities in all growth media; thus, they were selected and utilized for MBA wine fermentation. MBA wines co-fermented with non-*Saccharomyces* yeasts (*W. anomalus* JK04 or *M. pulcherrima* S36) and *S. cerevisiae* W-3 showed significantly increased levels of linalool, citronellol, and geraniol compared to MBA wine fermented with *S. cerevisiae* W-3 (control). In a sensory evaluation, the flavor, taste, and overall preference scores of the co-fermented wines were higher than those for the control wine, suggesting that *W. anomalus* JK04 and *M. pulcherrima* S36 are favorable wine starters for improving Korean MBA wine quality.

## 1. Introduction

The properties of yeast and its metabolites are affected by several environmental factors, which have long been utilized in various fields, such as enzymology, baking, alcoholic beverage production, and pharmacology [1]. In particular, *Saccharomyces cerevisiae*, the most well-known and well-studied yeast, is typically used for winemaking because it can survive under high ethanol concentrations, ferments faster than other strains, and secretes toxic compounds to inhibit the growth of undesirable microorganisms during fermentation [1,2]. Although *S. cerevisiae* has many advantages for alcohol fermentation, the potential application of various non-*Saccharomyces* yeasts has also been explored. These strains are widely found in nature and mainly grow during the initial step of wine fermentation, consequently affecting the taste and aroma of wine by producing volatile aromatic compounds such as esters, monoterpenes, higher alcohols, and acids [3,4,5]. For example, *Hanseniaspora guilliermondii* and *Wickerhamomyces anomalus* (formerly known as *Pichia anomala*) have been reported to exhibit acetate ester-forming activities, resulting in an acetate ester-enriched wine during fermentation by co-culture with *S. cerevisiae* [3,4,6]. *Issatchenkia terricola*, *Pichia kudriavzevii*, *W. anomalus*, *H. uvarum*, *Metschnikowia pulcherrima* and *Dekkera bruxellensis* also enhance the aromatic properties of wine by releasing β-glucosidase, which hydrolyzes the glucosidic bonds from various aglycone structures to monoterpenes, precursors of aromatic compounds [7,8,9,10]. Recently, some of the major starter companies have commercialized non-*Saccharomyces* yeasts for winemaking, thereby contributing to the improvement and diversification of commercial wine products [11].

Campbell Early and Muscat Bailey A (MBA) grapes are the most commonly cultivated varieties in Korean winemaking [12]. Campbell Early grapes contain high malic acid contents caused by early harvesting, which enhance the grape color. Chaptalization of Campbell Early grapes is required for winemaking due to their low sugar content (14–15 Brix) [13,14]. The MBA grape, a hybrid of Muscat Hamburg and Bailey cultivars, contains less acidity and a higher sugar content (18–21 Brix) compared to Campbell Early grapes, suggesting that MBA grapes are more suitable for winemaking as they do not require chaptalization [15]. Furthermore, Muscat grape varieties contain superior monoterpene alcohols, such as linalool, geraniol, nerol, citronellol, and α-terpineol [16,17,18]. A small increase in these compounds during fermentation can significantly affect the wine aroma due to the low odor thresholds [16,17,18]. Despite the potential of MBA grapes against volatile terpene compounds, Korean researchers have focused on the processing strategies irrelevant to wine aroma, such as freeze-concentration, pectinase treatment, blending with *Aronia*, and the reduction of acetaldehyde, methanol, and fusel oils [15,19,20,21]. Thus, the investigation of the ability of non-*Saccharomyces* yeasts to enhance volatile terpene compounds in MBA wine will contribute to increased competitiveness for Korean wine.

The objective of this study is to select non-*Saccharomyces* yeast isolates as candidate wine starters for improving the aroma of Korean MBA wine. Nine types of non-*Saccharomyces* yeasts were isolated from various Korean food materials, and their physiological and biochemical characteristics were investigated for comparison among isolates. In addition, the enzymatic characteristics and β-glucosidase activities of each strain were examined to evaluate the potential for aroma-forming activity. Finally, various fermentation characteristics and the volatile aromatic profiling of MBA wine fermented by co-culture with non-*Saccharomyces* yeasts and *S. cerevisiae* strain W-3 (as a control) were investigated.

## 2. Materials and Methods

### 2.1. Strains and Isolation

*Saccharomyces cerevisiae* W-3, an industrial wine yeast strain, was utilized as a control for all experiments. *Wickerhamomyces anomalus* JK04 and *Torulaspora delbrueckii* JK08, isolated from *nuruk* and stored in the laboratory, were used as non-*Saccharomyces* comparison strains [22]. Seven other strains of non-*Saccharomyces* yeasts were isolated from Sémillon grapes, persimmons, and MBA grapes cultivated in Korea. The non-*Saccharomyces* yeasts were isolated by crushing the food materials, followed by immersion in 0.9% NaCl solution. The samples were then sequentially diluted, and the aqueous solution was spread onto yeast peptone dextrose (YPD) agar plates and incubated at 30 °C for 2 days. All isolates were selected at random.

### 2.2. Polymerase Chain Reaction (PCR) and PCR-Restriction Fragment Length Polymorphism (RFLP)

Genomic DNA, used as the PCR template, was isolated from yeast cells grown in YPD medium for 24 h as described by Kaiser and Philippsen [23,24]. Internal transcribed spacer (ITS) region oligonucleotide primer sets (ITS1, 5′-CAT TTA GAG GAA CTA AAA GTC G-3′ and ITS4, 5′-CCT CCG CTT ATT GAT ATG C-3′), synthesized by Bioneer Co. (Chongwon, Korea), were used for PCR and PCR-RFLP analysis [25]. PCR was performed in a 50 µL reaction volume using TaKaRa Taq DNA polymerase (Takara Shuzo Co., Otsu, Japan) with a GENE cycler (Bio-Rad Co., Richmond, CA, USA). The PCR mixture consisted of 1 µg yeast genomic DNA, 10 pmol of each primer, 1 U Taq DNA polymerase, 0.25 mM of each dNTP, 10 mM Tris-HCl (pH 8.3), 50 mM KCl, and 2.5 mM MgCl_2_. The PCR cycle program for DNA amplification comprised one cycle at 94 °C for 3 min, 35 cycles of 94 °C for 45 s, 55 °C for 60 s, 72 °C for 60 s, and one cycle of 72 °C for 10 min. For PCR-RFLP, excess dNTPs and primers were removed from the PCR products using a PCR purification kit (Solgent, Daejeon, Korea). Then, appropriate amounts of amplicons were digested at 37 °C for 1 h with 0.5 µL of *Hinf* I, *Hae* III, and *Hpa* I endonucleases (Takara Shuzo Co., Otsu, Japan). DNA fragments were resolved on a 2.0% agarose gel according to standard methods [26], and a 100 bp Plus DNA Ladder (Solgent, Daegeon, Korea) was used as a marker to check the DNA size. To identify the isolated yeasts, the nucleotide sequences of the ITS I-5.8S-ITS II region were compared with those available in the GenBank database using the BLAST feature of the National Center for Biotechnology Information database http://blast.ncbi.nlm.nih.gov/Blast.cgi [27].

### 2.3. Enzymatic Activity and β-Glucosidase Activity

An API ZYM kit was used to analyze the enzymatic activities of the non-*Saccharomyces* yeasts. All strains were diluted to 5.0–6.0 McFarland standard with API suspension medium, inoculated into API ZYM test strips, and then incubated at 37 °C for 4 h. After the addition of one drop of ZYM reagent, reagents A and B from the kit were serially added to each strip and the strips were incubated for 5 min, allowing for colorimetric analysis [28]. Enzyme activity was classified from 0 (no activity) to 5 (maximum activity) using the API ZYM color reaction chart. Strains showing β-glucosidase activity were then further selected for culture in three kinds of media at 30 °C for 72 h to determine the effect of different types of carbon source on the β-glucosidase activity: YPD (1% yeast extract, 2% peptone, and 2% glucose), YPDC (1% yeast extract, 2% peptone, 1% glucose, and 1% cellobiose), and YPC (1% yeast extract, 2% peptone, and 2% cellobiose). Cells were collected every 12 h by centrifugation at 10,000× *g* for 3 min, and the enzyme activity of the supernatant was quantified. β-glucosidase activity was analyzed by measuring the amount of *p*-nitrophenyl-β-d-glucopyranoside (*p*NPG, Sigma, St. Louis, MO, USA) according to a slight modification of Swangkeaw’s method [8]. The centrifuged supernatant (0.1 mL) was mixed with 0.2 mL of 4 mM *p*NPG solution and 0.2 mL of 0.1 M citrate phosphate buffer (pH 5.0). The mixture was incubated at 30 °C for 30 min and the reaction was subsequently stopped by adding 2.0 mL of 2 M sodium carbonate. The *p*NPG released from this mixture was measured spectrophotometrically at 405 nm. One unit (U) of enzyme activity was defined as the amount of enzyme that released 1 μmol of *p*NPG per minute under the experimental conditions.

### 2.4. Physiological Characteristics

Carbon fermentation and assimilation were analyzed and compared according to a standard yeast identification method [29]. The ability of carbon fermentation was confirmed by carbon dioxide gas filled in Durham tubes of 5 mL liquid medium containing 4.5 g/L yeast extract, 7.5 g/L peptone, and 20 g/L sugars, including d-glucose, d-galactose, maltose, sucrose, lactose, and trehalose (except for raffinose, where 4% is commonly used because some strains use only part of this molecule). Each yeast was cultured at 25 °C for 14 days without shaking. The results were scored depending on the time taken to fill the insert with gas and the amount accumulating, as follows: +, strongly positive, insert filled within 7 days; d, delayed positive, insert rapidly filled, but only after more than 7 days; s, slowly positive, insert slowly filled after more than 7 days; w, weakly positive, the insert was not fully filled with gas (less than one-third full is often considered weak, whereas greater than one-third full is positive); −, negative, no accumulation of gas in the insert. To determine the ability of carbon assimilation, each yeast cell was cultured in 5 mL YNB broth (6.7 g/L yeast nitrogen base, and 20 g/L sugars except for raffinose, which was 4%) at 30 °C for 14 days. The 43 carbon compounds used in this study were as follows: hexoses (d-glucose, d-galactose, and l-sorbose), disaccharides (cellobiose, lactose, maltose, melibiose, sucrose, and trehalose), trisaccharides (melezitose and raffinose), polysaccharides (inulin and soluble starch), pentoses (d-arabinose, l-arabinose, d-ribose, l-rhamnose, and d-xylose), alcohols (erythritol, galactitol, d-glucitol, glycerol, myo-inositol, d-mannitol, ribitol, ethanol, methanol, xylitol, and l-arabinitol), organic acids (citrate, dl-lactate, succinate, and d-gluconate), glycosides (α-methyl-d-glucoside, salicin, and arbutin), and other compounds (d-glucosamine hydrochloride, *N*-acetyl-d-glucosamine, hexadecane, 2-keto-d-gluconate, 5-keto-d-gluconate, saccharate, and d-glucuronate). The results were presented as follows: +, positive, completely grown after 1 week; d, delayed positive, rapidly grown but only after 2 weeks; s, slow positive, slowly grown over 2 weeks; w, weak positive, slightly grown; −, negative, no growth.

### 2.5. Environmental Tolerance and Fermentation Rates

The strains’ tolerances to several environmental stresses, including glucose, ethanol, acid, and sulfur dioxide (SO_2_), were determined by culturing the isolates on modified YPD broth, in which the media were adjusted or supplemented to 20% and 50% glucose; 8% and 12% ethanol; pH 2 and 4; and 200 and 500 mg/L potassium metabisulfite (K_2_S_2_O_5_). The ethanol and SO_2_ solutions were filter-sterilized before addition to the YPD broth. Each strain was first cultured in 5 mL YPD at 30 °C for 16 h, then inoculated to the respective modified YPD broth with initial cell density of 0.5 (OD 600 nm) and incubated at 30 °C for 48 h. The pH was adjusted by the addition of hydrochloride (HCl) solution. Cell suspensions were quantified spectrophotometrically at 600 nm.

The fermentation rate was analyzed to compare the fermentability of each strain incubated in 100 mL YPD broth consisting of 20% glucose at 20 °C without shaking until the fermentation was complete. A water-trap apparatus containing concentrated H_2_SO_4_ was attached to the top of each flask to trap the water evaporating from the flask during fermentation. The amount of CO_2_ produced was indirectly measured as the decrease in the weight of the whole flask. The fermentation ratio was expressed as the percentage of the amount of CO_2_ produced according to the theoretical CO_2_ production from glucose due to ethanol fermentation [30]. The fermentation rate was calculated as follows:(1)$$\left(R_{CO2}\times \frac{M_{Glu}}{M_{CO2}}\right)\;\times \;\frac{1}{W_{Glu}}\;\times \;100$$
where *R_CO_*_2_ is the weight of reduced carbon dioxide during alcohol fermentation, *M_Glu_* and *M_CO_*_2_ are the molecular weights of glucose and carbon dioxide, respectively, and *W_Glu_* is the initial weight of the glucose in the YPD broth (20 g).

### 2.6. Wine Fermentation

Muscat Bailey A grapes, cultivated in Yeongcheon, Korea, were washed, stemmed, and crushed for wine fermentation. Grape must (5 kg) was fermented by co-culture of non-*Saccharomyces* yeasts (*W. anomalus* JK04 and *M. pulcherrima* S36) and *S. cerevisiae* W-3 (9:1, *v/v*) at 20 °C for 7 days, and 100 mg/L K_2_S_2_O_5_ was added to prevent bacterial contamination. The control wine was fermented by single-culture of *S. cerevisiae* W-3.

### 2.7. Analysis of Wine Characteristics and Viable Cell Counts

All the wine samples were prepared by centrifugation (3578× *g*, 10 min) to analyze their fermentation characteristics. Soluble solids were measured with a refractometer and reducing sugar content was analyzed using dinitrosalicylic acid according to AOAC guidelines [31]. The pH was measured with a pH meter (Mettler-Toledo, Schwerzenbach, Switzerland), and total acidity was determined by titration of filtrates with 0.1 N NaOH (expressed as % tartaric acid). Alcohol content was measured with a hydrometer based on the specific gravity of wine distillates (expressed as % *v*/*v*) at 15 °C [31]. The total phenolic compound content was determined by the Folin–Ciocalteau method [32]. The organic acid content was determined by HPLC (Model Prominence, Shimadzu, Kyoto, Japan) using a PL Hi-Plex H column (diam. 7.7 × 300 mm; Agilent Technologies, Santa Clara, CA, USA). The column chromatography conditions were as follows: flow rate, 1 mL/min; temperature, 65 °C; mobile phase, 0.005 mol sulfuric acid. Organic acids were detected with a refractive index detector (RID-10A, Shimadzu).

### 2.8. Volatile Aromatic Compounds in Muscat Bailey A Wine

To confirm the effects of non-*Saccharomyces* yeasts on wine aroma, the volatile aromatic compounds in MBA wine were quantified using gas chromatography mass spectrometry (7890A GC-MS; Agilent, Santa Clara, CA, USA) equipped with a flame ionization detector (FID). The separation was performed with a DB-WAX column (60 m × 250 μm × 0.25 mm; Waters, Milford, MA, USA), and detected with a triple-axis Agilent 5975C Inert XL MSD detector. Helium was used as a carrier gas with a constant flow rate of 1 mL/min. The oven of the chromatograph was programmed as follows: initial hold at 40 °C for 2 min, increased by 2 °C/min up to 220 °C, and then increased continuously at 20 °C/min up to 240 °C, with a final hold at 240 °C for 5 min. Volatile aromatic compounds were collected from the wine using a solid-phase microextraction (SPME) fiber (50/30 μm DVB/CAR/PDMS; Supelco, Bellefonte, PA, USA) in headspace (HS) mode with magnetic stirring. Five milliliters of the sample was placed in an HS vial (20 mm, PTFE/silicon septum, magnetic cap), and 1.25 g of NaCl was added to increase the volatile aromatic compound concentration in the headspace by increasing the retention of water-soluble components. Prior to extraction, the sample was shaken in a water bath at 35 °C for 20 min to achieve equilibrium. The SPME fiber was then spiked into the vial and exposed at 35 °C for 40 min. The commercial standards for quantification were supplied by Sigma Aldrich (St. Louis, MO, USA). Volatile aromatic compound identification was based on comparison of their gas chromatograph retention times and mass spectra with reference to spectral data from the Wiley9Nist 0.8 library (Wiley9Nist 0.8 library, mass spectral search program, v. 5.0, USA) [33]. The amount of each compound in the wine was calculated using the peak area based on the chemical standards.

### 2.9. Sensory Evaluation

Sensory evaluation was performed using the nine-point hedonic scale. Before the sensory evaluation, each wine was placed in a sample bottle and left undisturbed at room temperature for 1 h with the lid closed. After flavor evaluation, each wine was poured into wine glasses to evaluate the color, taste, and overall preference. The panel was composed of 20 judges with sensitive taste discrimination from the Department of Food Science and Technology, Kyungpook National University, Korea. Each judge evaluated the wines with at least 3-min intervals between samples, and water was provided to cleanse the palate. Sensory scores were assigned as follows: 9 (like extremely), 5 (neither like or dislike), 1 (dislike extremely).

### 2.10. Statistical Analysis

All experiments were carried out in at least triplicate and the results were analyzed using the Statistical Package for the Social Sciences (SPSS, v. 12.0 for Windows) to obtain means and standard deviations. Significance was determined at *p* < 0.05 using Student’s *t*-test and one-way analysis of variance, followed by Duncan’s multiple range test.

## 3. Results and Discussion

### 3.1. Isolation and Identification of Non-Saccharomyces Yeasts Using PCR-RFLP Analysis

Various non-*Saccharomyces* yeasts isolated from several food materials were tested for their potential in improving the aroma of wine during fermentation. *W. anomalus* JK04 and *T. delbrueckii* JK08 were previously isolated from *nuruk*, which is used as a source of microflora for rice wine and to improve the organoleptic properties in bread and wine [22,34]. In addition, 36 isolates from MBA grapes, 12 isolates from persimmons, and 36 isolates from Sémillon grapes were classified using PCR-RFLP analysis (Appendix A). Seven, four, and two types of strain were distinguished in Sémillon grapes, persimmons, and MBA grapes, respectively, depending on the fragment patterns digested with *Hinf* I, *Hae* III, and *Hpa* I endonucleases (Appendix A). The seven isolates showing the highest OD value among each of the species when cultured in sterilized MBA grape juice at 30 °C for 48 h were selected for further study and identified as *Starmerella bacillaris* MR35, *Candida quercitrusa* P6, *Pichia kluyveri* P11, *Hanseniaspora vineae* S7, *H. uvarum* S8, *Candida railenensis* S18, and *Metschnikowia pulcherrima* S36 using amplicons of the ITS I-5.8S-ITS II region in a BLAST search in the NCBI database (www.ncbi.nlm.nih.gov). The ITS I-5.8S-ITS II region DNA sequences of the nine non-*Saccharomyces* yeasts were submitted to the GenBank database under accession numbers MF574300–MF574308 (Table 1).

Several researchers have used PCR-RFLP analysis with different restriction endonucleases to confirm the dynamics of non-*Saccharomyces* yeasts during spontaneous wine fermentation or at the early stage of winemaking for rapid identification [36,37,38,39]. Alternatively, yeast dynamics can be investigated with slightly modified methods, such as simple sequence repeat fingerprinting, tandem repeat-tRNA PCR, or denaturing gradient gel electrophoresis PCR [40,41,42]. In this study, the results of PCR-RFLP analysis using different endonucleases quickly distinguished between each species with clearly different fragment patterns.

### 3.2. Physiological Characteristics of Non-Saccharomyces Yeasts

The carbon fermentation and assimilation analysis showed that all strains exhibited similar patterns to the reference strains described by Kurtzman’s method [29] (Table 2). Among nine non-*Saccharomyces* yeasts, *P. kluyveri* P11, *H. vineae* S7, *H. uvarum* S8, and *M. pulcherrima* S36 utilized only glucose as a carbon source for assimilation, which may result in low ethanol contents after fermentation or a slow fermentation rate. Accordingly, many researchers have applied these yeasts in wine fermentation using co-fermentation with *S. cerevisiae* to increase thiol concentration [43], improve the organoleptic quality and reduce the volatile acidity [44], increase flavor diversity [45], and increase foam persistence (especially in sparkling wine) and change aromatic profile [46]. *W. anomalus* JK04 and *T. delbrueckii* JK08 showed similar carbon fermentation patterns to the control strain *S. cerevisiae* W-3, except for galactose (delayed positive) and maltose (negative), respectively. *S. bacillaris* MR35 could also use sucrose and raffinose as carbon sources for fermentation, whereas *C. quercitrusa* P6 and *C. railenensis* S18 could also utilize galactose. *W. anomalus* JK04, *T. delbrueckii* JK08, *C. quercitrusa* P6, *C. railenensis* S18, and *M. pulcherrima* S36 were able to use more types of carbon sources for assimilation than *S. cerevisiae* W-3, whereas *S. bacillaris* MR35 showed positive results for only glucose, sucrose, and raffinose (matching the positive results in carbon fermentation analysis), with a weak positive result for L-sorbose, and negative results for all other carbon sources. In addition, *P. kluyveri* P11, *H. vineae* S7, and *H. uvarum* S8 also utilized only a few types of carbon source for assimilation. *W. anomalus* JK04, *H. vineae* S7, *H. uvarum* S8, *C. railenensis* S18, and *M. pulcherrima* S36 showed positive results for the assimilation of cellobiose, which is a known substrate of β-glucosidase and is involved in the production of monoterpenes related to the aromatic compounds in wine. Indeed, these strains have been used in investigations for increasing the monoterpene content to improve wine aroma [5,8].

### 3.3. Biochemical Characteristics for Carbon Assimilation of Non-Saccharomyces Yeasts

The tolerances to several environmental parameters were investigated in the non-*Saccharomyces* strains to determine their potential in winemaking and aroma improvement (Table 3). Although only minor growth inhibition was detected between YPD broth containing 50% glucose and 20% glucose for all strains, there was still sufficiently high cell growth in both conditions, indicating that osmotic stress has a negligible influence on wine fermentation. In contrast, all of the non-*Saccharomyces* yeasts, except for *W. anomalus* JK04, showed significantly reduced cell growth in YPD broth containing 8% ethanol, and *W. anomalus* JK04 also showed low cell growth in YPD broth containing 12% ethanol, supporting the presence of non-*Saccharomyces* yeasts during the initial phase of wine fermentation, thereby exerting a greater influence on the formation of various aromatic compounds [47,48]. In the pH tolerance analysis, *Hanseniaspora* sp. (S7 and S8) and *M. pulcherrima* S36 showed weak pH tolerances when cultured in YPD broth adjusted to pH 4, which is similar to the acidic condition of wine. Moreover, *W. anomalus* JK04, *S. bacillaris* MR35, *C. quercitrusa* P6, *H. uvarum* S8, and *C. railenensis* S18 showed strong acid tolerances when cultured in YPD broth adjusted to pH 2. *Hanseniaspora* spp. (S7 and S8) and *S. bacillaris* MR35 were very sensitive to SO_2_ stress conditions, indicating that lower SO_2_ treatment is preferable for wine fermentation by *S. bacillaris* MR35 or *Hanseniaspora* species. The results of fermentation rate indicated that *W. anomalus* JK04, *T. delbrueckii* JK08, *S. bacillaris* MR35, and *H. vineae* S7 converted over 80% of the glucose to CO_2_, whereas the other non-*Saccharomyces* strains showed relatively low fermentation rates (Table 3). Because the cultivation times of non-*Saccharomyces* yeasts were too long to complete fermentation, these non-*Saccharomyces* yeasts are most likely to affect wine properties during the early phase of fermentation (Appendix A). Although some strains showed good potential for sole use in winemaking because they fermented over 80% of the glucose available, co-fermentation with *S. cerevisiae* and non-*Saccharomyces* yeasts is still considered a better approach for producing more favorable wines and avoiding contamination of foreign microorganisms by reducing the fermentation time.

### 3.4. Enzymatic Activity and the Effect of Growth Medium on β-Glucosidase Production

The enzymes produced by non-*Saccharomyces* yeasts play an important role in the development of abundant wine aroma and improving the sensory properties of wine [49]. Nineteen types of enzyme activities of non-*Saccharomyces* yeasts were verified using API ZYM (Figure 1). None of the non-*Saccharomyces* strains showed lipase, trypsin, α-chymotrypsin, β-galactosidase, β-glucuronidase, *N*-acetyl-β-glucosaminidase, α-mannosidase, or α-fucosidase activities, whereas esterase lipase, leucine arylamidase, valine arylamidase, cystine arylamidase, and naphthol-AS-BI-phosphohydrolase activities were detected in all strains (except for crystine arylamidase in *H. uvarum* S8). Alkaline phosphatase activity was detected weakly or not at all in all strains, whereas acid phosphatase activity was detected in all strains, although *H. vineae* S7, *H. uvarum* S8, and *M. pulcherrima* S36 exhibited relatively weaker activity compared to that of the other strains, which is in line with the result of the acid tolerance analysis (pH 4) (Table 3). α-Glucosidase activity was detected in *S. cerevisiae* W-3, *W. anomalus* JK04, *C. quersitrusa* P6, *H. vineae* S7, *C. railenensis* S18, and *M. pulcherrima* S36, whereas α-galactosidase activity was detected only in *T. delbrueckii* JK08, in accordance with the results of maltose and melibiose assimilation analysis, respectively. Esterase activity, which is involved in the formation of volatile aromatic ester compounds [50], was detected at a similar level or weakly in all non-*Saccharomyces* yeasts compared to that of the control (*S. cerevisiae* W-3). β-Glucosidase activity, which is related to the release of terpenes into wine [51], was detected in *W. anomalus* JK04, *H. vineae* S7, *H. uvarum* S8, *C. railenensis* S18, and *M. pulcherrima* S36, corresponding to the results of the cellobiose assimilation analysis (Figure 1). The five strains with β-glucosidase activity were selected for further analysis.

The β-glucosidase activity was quantified in five non-*Saccharomyce*s strains grown on media with different carbon sources: YPD, YPDC, and YPC (Figure 2). The β-glucosidase production of *W. anomalus* JK04 and *M. pulcherrima* S36 was higher than that of the other strains, regardless of the growth medium composition. The β-glucosidase activities of *W. anomalus* JK04 and *M. pulcherrima* S36 were 1.72, 2.03, and 2.55 U mL^−1^ min^−1^ (JK04) and 2.06, 2.76, and 2.55 U mL^−1^ min^−1^ (S36) when cultured in YPD, YPDC, and YPC, respectively. Although *H. vineae* S7 also showed increased β-glucosidase activity in all conditions, it was still relatively weaker compared to that of *W. anomalus* JK04 and *M. pulcherrima* S36 (1.03–1.64 U mL^−1^ min^−1^). In the case of *C. railenensis* S18, significantly high β-glucosidase activities were detected when it was cultured in YPD and YPDC (2.00 and 1.75 U mL^−1^ min^−1^, respectively), whereas limited activity was detected when it was cultured in YPC. *H. uvarum* S8 showed very weak β-glucosidase activities in all conditions, which differed to the result of the API ZYM kit (Figure 1). Among those five strains, *W. anomalus* JK04 and *M. pulcherrima* S36 showed relatively high β-glucosidase activities in all conditions. β-Glucosidase is a common enzyme that exists in nature, but β-glucosidases produced by *S. cerevisiae* and grapes are typically inactivated or inhibited under wine production conditions, such as pH 3–4 or high sugar levels [5,52,53]. Thus, several researchers have studied non-*Saccharomyces* yeasts, including *H. uvarum*, *W. anomalus*, *P. kudriavzevii*, *M. pulcherrima*, and *Issatchenkia terricola*, as β-glucosidase producers for improving wine aroma with high β-glucosidase activities and without glucose and low-pH repression [5,10,52,54]. In the present study, *W. anomalus* JK04 and *M. pulcherrima* S36, which showed the highest β-glucosidase activities, were selected for further investigation of volatile terpene compounds.

### 3.5. Effect of β-Glucosidase-Producing Non-Saccharomyces Yeasts on Fermentation Characteristics and Volatile Aromatic Profiling of MBA Wine

The fermentation characteristics of MBA wine co-fermented with *W. anomalus* JK04 (JK04 wine) or *M. pulcherrima* S36 (S36 wine), which showed the highest β-glucosidase producing activities, and *S. cerevisiae* W-3 were investigated (Figure 3 and Table 4). The viable cell count of *W. anomalus* JK04 was maintained at > log 6 CFU/mL until fermentation was complete, whereas the viable cell count of *M. pulcherrima* S36 dramatically decreased after the third day of fermentation and no viable cells were detected when fermentation was completed. The alcohol content of the S36 wine was 1% lower than that of the other MBA wines, probably due to the lower fermentation rate of *M. pulcherrima* S36 compared to that of *S. cerevisiae* W-3 and *W. anomalus* JK04 (Table 3). The pH, reducing sugars, and total phenolic compound contents were not significantly different between the wines, whereas the total acidities of the co-fermented wines were slightly increased. The results of the organic acid analysis showed that the acetic acid content of the co-fermented wines was slightly increased, whereas other compounds were not significantly different. In the volatile ester compound analysis, a total of 13 kinds of volatile ester compounds were detected. The JK04 wine showed higher ethyl acetate and isobutyl acetate contents compared to the control wine, and the S36 wine had increased isobutyl acetate, isobutyl octanoate, ethyl decanoate, ethyl octanoate, isoamyl octanoate, and methyl salicylate contents compared to the other wines, whereas the highest amounts of n-hexyl acetate and 2-phenylethyl acetate were detected in the control wine. Of note, all the wines had significantly higher isoamyl acetate and 2-phenylethyl acetate contents compared with those generally reported in other studies. Regarding volatile terpene compounds, only three kinds of volatile terpene compounds were detected. The linalool, citronellol, and geraniol contents of the JK04 and S36 wines were significantly increased to 13.8–46.9% and 16.5–27.2%, respectively, which were higher than those of the control wine. In the sensory evaluation, the co-fermented wines received higher scores for flavor, taste, and overall preference than the control wine.

Gonzalez et al. (2013) suggested that using Crabtree-negative non-*Saccharomyces* yeasts (*M. pulcherrmia* or *W. anomalus*) with *S. cerevisiae* could result in low ethanol contents in final wines [55]. *M. pulcherrima* S36 could be also considered as a potential wine starter for producing low-alcohol wine. In contrast with *M. pulcherrima* S36, *W. anomalus* JK04 produced a similar ethanol content to the control wine, but this strain could survive under the high ethanol concentration due to its high ethanol tolerance (Table 3), indicating that *W. anomalus* JK04 can affect the wine quality across the fermentation process. Similarly, β-glucosidase from *P. anomala* MDD24 has also been reported to be very efficient at releasing aromatic compounds during the final phase of wine fermentation due to its high ethanol tolerance [8,54]. While *W. anomalus* and *M. pulcherrima* have been known to produce ethyl acetate and fruity esters (especially ethyl octanoate) during fermentation, respectively [11,56], several researchers have also paid attention to applying the β-glucosidase activities of these strains to improve the volatile terpene compounds in wine [7,8,10,51,54,57]. The formation of volatile aromatic compounds, including monoterpenes, is influenced by many factors, such as grape cultivars, geological differences, soil, climate, water, nitrogen fertilization, and harvest time [58,59]. Although only three kinds of volatile terpene compounds, i.e., linalool, citronellol, and geraniol, were detected in this study, the increase of these compounds levels due to *W. anomalus* JK04 and *M. pulcherrima* S36 resulted in a significant influence on the odor perception of MBA wine because of the compounds’ very low odor thresholds (0.025, 0.1, and 0.02 mg/L, respectively) [60]. Moreover, all the wines had extraordinarily higher amount of some volatile ester compounds such as isoamyl acetate and 2-phenylethyl acetate than that generally reported [61]. Given our previous studies that used single or co-fermentation with *S. cerevisiae* W-3 and non-*Saccharomyces* yeasts, production of excessive isoamyl acetate and 2-phenylethyl acetate is a distinctive trait of *S. cerevisiae* W-3 [2,62]. Some non-*Saccharomyces* yeasts, including *M. pulcherrima*, have been reported to improve wine color owing to their ability to adsorb grape anthocyanins in their cell walls during fermentation [63], indicating that the S36 wine receives a higher color score than other wines. A relatively low ethanol content of the S36 wine also affects the taste of MBA wine.

## 4. Conclusions

In this study, β-glucosidase-producing non-*Saccharomyces* yeasts were investigated to improve Korean MBA wine quality by increasing volatile terpene compounds. A number of non-Saccharomyces yeasts were isolated from various food materials and their physiological, biochemical, and enzymatic properties were investigated. A total of five species of non-*Saccharomyces* yeasts were confirmed to have β-glucosidase activities. Among these strains, *W. anomalus* JK04 and *M. pulcherrima* S36 had the highest β-glucosidase activities in all media conditions with different carbon sources. The MBA wines co-fermented with these two non-*Saccharomyces* yeasts and *S. cerevisiae* W-3 had significantly increased linalool, citronellol, and geraniol contents. In a sensory evaluation, the flavor, taste, and overall preference scores of these two co-fermented wines were higher than those of the control wine. In conclusion, *W. anomalus* JK04 and *M. pulcherrima* S36 can contribute to the improved quality of Korean MBA wine.

## Figures and Tables

**Figure 1 microorganisms-08-01552-f001:**
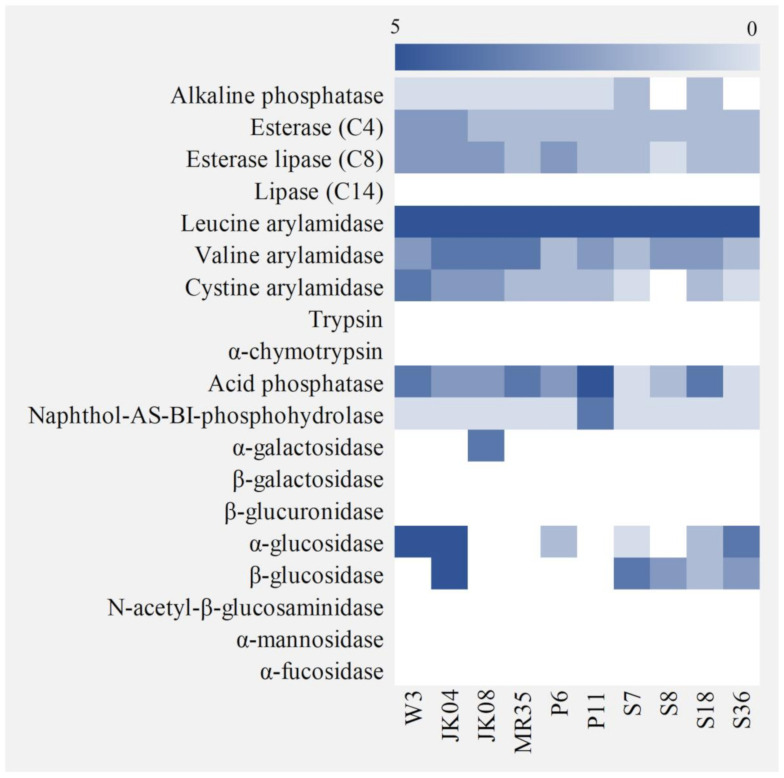
Heatmap of enzymatic activities of *Saccharomyces cerevisiae* W-3 and non-*Saccharomyces* yeasts isolated from various food materials. All activities were determined using an API ZYM kit. W-3, *Saccharomyces cerevisiae* W-3; JK04, *Wickerhamomyces anomalus* JK04; JK08, *Torulaspora delbrueckii* JK08; MR35, *Starmerella bacillaris* MR35; P6, *Candida quercitrusa* P6; P11, *Pichia kluyveri* P11; S7, *Hanseniaspora vineae* S7; S8, *H. uvarum* S8; S18, *Candida railenensis* S18; S36, *Metschnikowia pulcherrima* S36.

**Figure 2 microorganisms-08-01552-f002:**
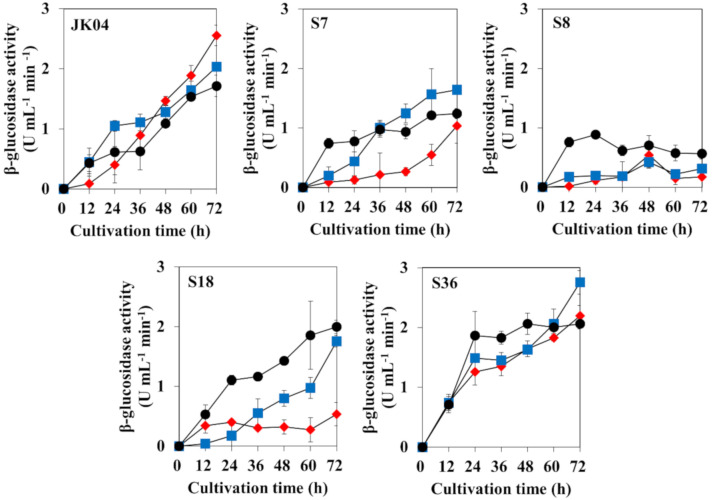
Effect of growth medium on β-glucosidase activities of five non-*Saccharomyces* yeasts when cultured on yeast peptone dextrose (YPD) (black circles), yeast peptone dextrose cellobiose (YPDC) (blue squares), and yeast peptone cellobiose (YPC) (red diamonds). JK04, *Wickerhamomyces anomalus* JK04; S7, *Hanseniaspora vineae* S7; S8, *H. uvarum* S8; S18, *Candida railenensis* S18; S36, *Metschnikowia pulcherrima* S36. YPD and YPC contain 2% of glucose and cellobiose, respectively. YPDC contains 1% of glucose and 1% of cellobiose.

**Figure 3 microorganisms-08-01552-f003:**
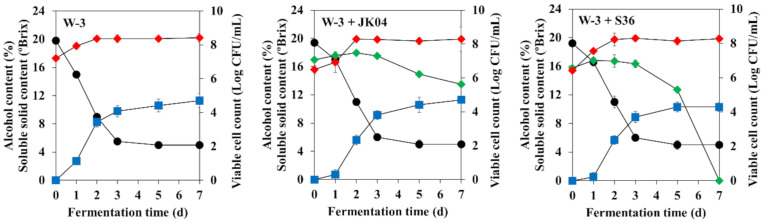
Soluble solids (black circles), alcohol contents (blue squares) and viable cell counts (diamonds) of Muscat Bailey A wine fermented by single culture of *Saccharomyces cerevisiae* W-3 (W-3), co-culture of *S. cerevisiae* W-3 and *Wickerhamomyces anomalus* JK04 (W-3 + JK04), and co-culture of *S. cerevisiae* W-3 and *Metschnikowia pulcherrima* S36 (W-3 + S36). Red and green symbols indicate viable cell counts of *S. cerevisiae* W-3 and non-*Saccharomyces* yeasts, respectively.

**Table 1 microorganisms-08-01552-t001:** Strains used in this study.

Species	Strains	Origin	GenBank Accession Number	Reference
*Saccharomyces cerevisiae*	W-3	-	-	[35]
*Wickerhamomyces anomalus*	JK04	Nuruk	MF574300	[22]
*Torulaspora delbrueckii*	JK08	Nuruk	MF574301	[22]
*Starmerella bacillaris*	MR35	Muscat Bailey A(MBA) grape	MF574302	This study
*Candida quercitrusa*	P6	Persimmon	MF574303	This study
*Pichia kluyveri*	P11	Persimmon	MF574304	This study
*Hanseniaspora vineae*	S7	Sémillon	MF574305	This study
*Hanseniaspora uvarum*	S8	Sémillon	MF574306	This study
*Candida railenensis*	S18	Sémillon	MF574307	This study
*Metschnikowia pulcherrima*	S36	Sémillon	MF574308	This study

**Table 2 microorganisms-08-01552-t002:** Carbon fermentation and assimilation characteristics of *Saccharomyces cerevisiae* W-3 and nine non-*Saccharomyces* yeasts.

Carbon Source	Strains										Carbon Source	Strains									
	W-3	JK04	JK08	MR35	P6	P11	S7	S8	S18	S36		W-3	JK04	JK08	MR35	P6	P11	S7	S8	S18	S36
**Fermentation**											**d-Xylose**	-	+	-	-	+	-	-	-	+	+
**d-Glucose**	+	+	+	+	+	+	+	+	+	+	**Erythritol**	-	+	-	-	-	-	-	-	-	-
**d-Galactose**	+	d	+	-	+	-	-	-	+	**-**	**Galactitol**	-	-	-	-	-	-	-	-	-	-
**Maltose**	+	+	-	-	-	-	-	-	-	-	**d-Glucitol**	-	+	+	-	+	-	-	-	+	+
**Sucrose**	+	+	+	+	-	-	-	-	-	-	**Glycerol**	-	+	-	-	+	+	-	-	+	+
**Lactose**	-	-	-	-	-	-	-	-	-	-	**myo-Inositol**	-	-	-	-	-	-	-	-	-	-
**Trehalose**	-	-	-	-	-	-	-	-	-	-	**d-Mannitol**	-	+	+	-	+	-	-	-	+	+
**Raffinose**	+	+	+	+	-	-	-	-	-	-	**Ribitol**	-	-	-	-	+	-	-	-	+	+
**Assimilation**											**Ethanol**	+	+	+	-	+	+	-	-	+	+
**d-Glucose**	+	+	+	+	+	+	+	+	+	+	**Methanol**	-	-	-	-	-	-	-	-	-	-
**d-Galactose**	+	+	+	-	+	-	-	-	+	+	**Citrate**	-	-	-	-	+	-	-	-	+	+
**l-Sorbose**	-	-	-	w	+	-	-	-	+	+	**dl-Lactate**	-	-	-	-	+	-	-	-	+	-
**Cellobiose**	-	+	-	-	-	-	+	+	+	+	**Succinate**	-	+	-	-	+	s	-	-	+	+
**Lactose**	-	-	-	-	-	-	-	-	-	-	**d-Gluconate**	-	-	-	-	+	s	-	+	w	w
**Maltose**	+	+	-	-	+	-	w	-	+	+	**α-Methyl-d-glucoside**	-	+	-	-	+	-	-	-	+	+
**Melibiose**	-	-	+	-	-	-	-	-	-	-	**Salicin**	-	+	-	-	-	+	+	+	+	+
**Sucrose**	+	+	+	+	+	w	-	-	+	+	**d-Glucosamine hydrochloride**	-	-	-	-	-	s	-	-	-	-
**Trehalose**	+	w	s	-	+	-	-	-	+	+	***N*-Acetyl-d-glucosamine**	-	-	-	-	+	+	-	-	+	+
**Melezitose**	+	+	w	-	+	-	-	-	+	+	**Hexadecane**	-	-	-	-	-	-	-	-	w	-
**Raffinose**	+	+	+	+	-	-	-	-	-	-	**2-Keto-d-gluconate**	+	-	+	-	+	-	-	+	+	+
**Inulin**	-	-	-	-	-	-	-	-	-	-	**5-Keto-d-gluconate**	+	+	+	-	+	+	-	-	+	+
**Starch**	-	+	-	-	-	-	-	-	-	-	**Saccharate**	-	-	-	-	-	-	-	-	-	-
**d-Arabinose**	-	-	-	-	-	-	-	-	d	-	**Arbutin**	-	+	-	-	d	-	+	+	+	+
**l-Arabinose**	-	-	-	-	-	-	-	-	-	-	**d-Glucuronate**	-	-	-	-	-	-	-	-	-	-
**d-Ribose**	-	-	-	-	-	-	-	-	-	-	**Xylitol**	-	w	-	-	+	+	-	-	+	+
**l-Rhamnose**	-	-	-	-	-	-	-	-	-	-	**l-Arabinitol**	-	-	-	-	-	-	-	-	-	-

* +, positive; -, negative; w, weak positive; d, delayed positive; s, slowly positive.

**Table 3 microorganisms-08-01552-t003:** Tolerance to various environmental stressors and fermentation rates of *Saccharomyces cerevisiae* W-3 and nine non-*Saccharomyces* yeasts.

Strains	Environmental Tolerance (OD 600 nm)								Fermentation Rate (%)
	20% Glucose	50% Glucose	8% EtOH	12% EtOH	pH 4	pH 2	200 mg/L K_2_S_5_O_2_	500 mg/L K_2_S_5_O_2_	
**W-3**	42.63 ± 1.24	32.70 ± 2.28	12.21 ± 1.64	3.00 ± 0.53	33.60 ± 1.13	1.16 ± 0.05	18.68 ± 2.11	20.78 ± 4.63	90.47 ± 1.30
**JK04**	28.13 ± 1.66	30.08 ± 1.52	15.20 ± 2.03	1.78 ± 0.66	23.63 ± 2.46	7.15 ± 0.47	27.33 ± 6.47	24.13 ± 1.60	87.40 ± 1.58
**JK08**	43.03 ± 2.51	33.73 ± 3.39	3.51 ± 0.46	1.22 ± 0.07	35.70 ± 2.85	1.17 ± 0.07	20.23 ± 3.57	16.98 ± 1.66	86.89 ± 0.36
**MR35**	38.80 ± 1.91	25.93 ± 2.81	2.30 ± 0.12	1.03 ± 0.42	25.48 ± 2.18	10.83 ± 1.00	15.69 ± 1.52	14.00 ± 1.93	84.12 ± 2.21
**P6**	24.30 ± 1.28	18.45 ± 1.49	2.14 ± 0.24	0.70 ± 0.03	30.73 ± 2.16	9.60 ± 0.79	30.83 ± 3.15	28.20 ± 2.21	53.32 ± 0.84
**P11**	28.85 ± 2.64	26.13 ± 2.72	1.43 ± 0.07	0.41 ± 0.01	24.60 ± 1.14	1.41 ± 0.21	20.38 ± 2.32	21.80 ± 3.18	37.35 ± 2.12
**S7**	31.75 ± 2.26	25.13 ± 1.39	1.25 ± 0.23	0.41 ± 0.04	16.03 ± 1.17	1.39 ± 0.08	16.75 ± 1.78	12.43 ± 1.44	87.50 ± 0.95
**S8**	28.40 ± 1.41	23.35 ± 3.75	1.12 ± 0.44	0.49 ± 0.02	10.21 ± 1.85	4.35 ± 0.92	12.12 ± 1.53	11.42 ± 1.25	41.55 ± 1.20
**S18**	25.68 ± 1.45	23.73 ± 2.65	1.15 ± 0.48	0.37 ± 0.03	25.93 ± 3.04	5.73 ± 0.25	34.18 ± 4.77	29.73 ± 3.53	49.74 ± 2.54
**S36**	29.10 ± 3.32	27.65 ± 1.21	0.78 ± 0.03	0.38 ± 0.02	8.05 ± 0.38	1.79 ± 0.24	26.35 ± 1.48	30.20 ± 4.99	44.52 ± 1.31

**Table 4 microorganisms-08-01552-t004:** Physicochemical properties and sensory scores of MBA wine fermented with co-cultures of *S. cerevisiae* W-3 (W-3) and non-*Saccharomyces* yeasts.

Property	Strains		
	W-3	W-3 + JK04	W-3 + S36
**Alcohol (%, *v*/*v*)**	11.3 ± 0.1a	11.3 ± 0.1a	10.3 ± 0.1b
**Soluble solid (°Brix)**	5.0 ± 0.1a	5.0 ± 0.0a	5.0 ± 0.1a
**pH**	3.73 ± 0.02a	3.78 ± 0.02a	3.77 ± 0.02a
**Total acidity (%)**	0.47 ± 0.00b	0.55 ± 0.01a	0.57 ± 0.03a
**Reducing sugar (%)**	0.12 ± 0.01a	0.12 ± 0.01a	0.12 ± 0.01a
**Total phenolic compounds (%)**	0.14 ± 0.01a	0.13 ± 0.01a	0.13 ± 0.01a
**Organic acids (mg/mL)**			
**Citric acid**	1.07 ± 0.11a	1.14 ± 0.13a	1.09 ± 0.10a
**Tartaric acid**	1.58 ± 0.10a	1.56 ± 0.16a	1.72 ± 0.13a
**Malic acid**	4.60 ± 0.26a	4.97 ± 0.32a	4.82 ± 0.24a
**Succinic acid**	1.67 ± 0.07a	1.55 ± 0.05a	1.62 ± 0.09a
**Acetic acid**	0.15 ± 0.01b	0.26 ± 0.03a	0.21 ± 0.02a
**Volatile ester compounds (mg/L)**			
**Ethyl acetate**	141.43 ± 12.38b	261.12 ± 23.05a	151.05 ± 19.67b
**Isobutyl acetate**	8.84 ± 0.95b	11.00 ± 0.68a	12.68 ± 1.04a
**Isoamyl acetate**	545.91 ± 41.94a	493.13 ± 36.86a	548.54 ± 46.25a
**Ethyl hexanoate**	153.47 ± 10.12b	131.47 ± 12.31b	182.65 ± 16.00a
**n-Hexyl acetate**	7.64 ± 0.35a	3.64 ± 0.21c	5.31 ± 0.38b
**Ethyl heptanoate**	1.20 ± 0.08a	1.20 ± 0.10a	1.41 ± 0.13a
**Ethyl octanoate**	426.41 ± 35.31b	430.52 ± 38.14b	513.40 ± 21.06a
**Isobutyl octanoate**	1.49 ± 0.11b	1.68 ± 0.13b	2.28 ± 0.20a
**Ethyl decanoate**	427.18 ± 31.68b	469.09 ± 29.57ab	534.14 ± 39.63a
**Isoamyl octanoate**	13.50 ± 1.65b	12.63 ± 1.21b	16.53 ± 1.34a
**Methyl salicylate**	31.60 ± 2.15b	32.54 ± 3.08b	39.06 ± 3.36a
**2-Phenylethyl acetate**	87.26 ± 6.53a	36.13 ± 2.62c	62.22 ± 4.49b
**Ethyl dodecanoate**	162.81 ± 12.36a	143.79 ± 10.30a	142.20 ± 11.27a
**Volatile terpene compounds (mg/L)/*Increase level (%)**			
**Linalool**	0.81 ± 0.07c	1.19 ± 0.06a / 46.9%	1.03 ± 0.08b / 27.2%
**Citronellol**	5.60 ± 0.25b	6.37 ± 0.30a / 13.8%	6.77 ± 0.28a / 20.9%
**Geraniol**	2.00 ± 0.13b	2.38 ± 0.11a / 19.0%	2.33 ± 0.13a / 16.5%
**Sensory score**			
**Color**	5.80 ± 1.82a	5.85 ± 1.95a	6.15 ± 1.84a
**Flavor**	5.20 ± 1.89a	5.60 ± 1.79a	5.45 ± 2.01a
**Taste**	4.15 ± 1.58a	4.50 ± 1.66a	4.65 ± 1.38a
**Overall preference**	5.65 ± 1.66a	6.30 ± 1.62a	6.20 ± 1.63a

***** a–c Different letters within the same row indicate a statistically significant difference (*p* < 0.05). ***** Increase in the levels of volatile terpene compounds in co-fermented wines were calculated by comparing with those of the control wine.

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
