# Peer review of "Isolation and Investigation of Potential Non-Saccharomyces Yeasts to Improve the Volatile Terpene Compounds in Korean Muscat Bailey A Wine"

_microorganisms, 2020, doi:10.3390/microorganisms8101552_

Round 1

Reviewer 1 Report

This study details the isolation of several yeast species from distinct environmental sources. Selected non-Saccharomyces strains are subsequently screened for their physiological and enzymatic properties to identify those with greater potential to improve aromatic compound production during the early stages of wine fermentation.
The authors apply a series of experimental procedures that are appropriate within the proposed scope of the manuscript. Some sections require a greater level of scientific detail.

Comments:

Line 80 - Replace types by strains.

Line 231 - Seven isolates were selected based on their respective growth rates. Please indicate which growth conditions were used for this initial screening.

Line 252 - Please specify the traits that are usually improved by the use of these yeasts during co-fermentation with S. cerevisiae.

Line 269 onward - This section details a screening for tolerance to some environmental stresses typically observed during fermentation. This screening was performed based on OD measurements after 48h growth in each condition. Firstly, it appears that the inoculation in each modified YPD medium was performed by a % dilution of a pre-culture and not to a specific OD 600. A footnote in table 3 states that the initial OD was approximately 0.5. Was this actually quantified or simply extrapolated? I am concerned that each strain may have grown to different extents in the pre-culture, which could be reflected in distinct initial cell densities for these experiments. Additionally, this experiment would benefit from CFU analysis. The use of different yeast species makes it likely that the same OD actually corresponds to different cell numbers. A control to determine the initial CFU count would be key to ensure each species is grown with the same cell concentration so that the final results are not biased due to different initial population sizes, which can impact stress tolerance.

Lines 290-291 - This appears to be structured as a conclusion. I believe the authors wish to convey that since these fermentations take a long time, if these strains have an effect in wine properties, it is most likely to occur during the early phase of fermentation.

Lines 292-294 - "Although some strains showed good potential for wine-making through single co-fermentation with S. cerevisiae". I do not believe co-fermentations have been performed at this point in the manuscript?
"non-Saccharomyces yeasts are still considered better for producing more favorable wines and avoiding contamination of foreign microorganisms by reducing the fermentation time". Please detail some of the features referred to as "favorable wines". More importantly, as the authors state above, non-Saccharomyces yeasts typically take a long time to finish the fermentation process and produce small amounts of ethanol. Stating that these yeasts are considered better to avoid contamination and reduce fermentation time seems contradictory. If the authors mean that S. cerevisiae co-fermentations with non-Saccharomyces species contribute to enhance these properties vs only S. cerevisiae fermentations, they should reference studies supporting such claims.

Figure 1 -  Units for the 0 to 5 scale in Figure 1 are required.

Section 3.5 - More discussion on the attained concentrations of produced metabolites is warranted. For example, the concentrations of isoamyl acetate or 2-phenylethyl acetate present in Table 4 are significantly higher than reported in other studies. Possible factors such as distinct fermentation conditions, medium composition, S. cerevisiae strains used, etc., that may justify the data should be discussed.

Author Response

Dear Reviewer 1,

We appreciate your kind review for improving the manuscript.

We have tried to solve your concerns and all the modifications in the manuscript indicated in red font with comments. Following are our detail responses against your comments.

Line 80 - Replace types by strains.

→ We revised it, thanks.

Line 231 - Seven isolates were selected based on their respective growth rates. Please indicate which growth conditions were used for this initial screening.

→ We added growth conditions for the primary selection in section 3.1 and supplementary Table S1, thanks.

Line 252 - Please specify the traits that are usually improved by the use of these yeasts during co-fermentation with S. cerevisiae.

→ We specified the traits by these yeasts during co-fermentation with S. cerevieiae, thanks.

Line 269 onward - This section details a screening for tolerance to some environmental stresses typically observed during fermentation. This screening was performed based on OD measurements after 48h growth in each condition. Firstly, it appears that the inoculation in each modified YPD medium was performed by a % dilution of a pre-culture and not to a specific OD 600. A footnote in table 3 states that the initial OD was approximately 0.5. Was this actually quantified or simply extrapolated? I am concerned that each strain may have grown to different extents in the pre-culture, which could be reflected in distinct initial cell densities for these experiments. Additionally, this experiment would benefit from CFU analysis. The use of different yeast species makes it likely that the same OD actually corresponds to different cell numbers. A control to determine the initial CFU count would be key to ensure each species is grown with the same cell concentration so that the final results are not biased due to different initial population sizes, which can impact stress tolerance.

→ Thanks to point out this. It was our mistake to describe the wrong initial inoculation in Material and Method section. Initial cell density for this experiment was 0.5 at OD 600 nm. Thus, we revised it, thanks.

→ In addition, we agree that the OD values won’t be able to reflect the corresponding cell numbers of different species and compare the cell growths of different species. In this study, however, we investigated the environmental tolerances as fundamental characteristics to figure out how much the strains can grow under the fermentation conditions (low pH, high sugar, high ethanol, and high SO2). Because the promising candidates of MBA wine starters were selected by enzymatic analysis, we don’t need to compare the environmental tolerances among different species. We know CFU can be more flexible to express the results. But re-investigating the environmental tolerances should take a very long time so we won’t be able to meet the deadline. We would think the OD value of each tolerance could also explain relative differences in same species regardless of different cell size. Some researchers also described the environmental tolerances using the OD values in their papers (see the links). So, we removed the statistical marks in Table 3 and revised some sentences in Section 3.3. Please kindly consider it, thanks.

https://www.sciencedirect.com/science/article/pii/S0023643820304217

https://www.sciencedirect.com/science/article/pii/S0963996918302126

https://www.sciencedirect.com/science/article/pii/S0168160509000476

Lines 290-291 - This appears to be structured as a conclusion. I believe the authors wish to convey that since these fermentations take a long time, if these strains have an effect in wine properties, it is most likely to occur during the early phase of fermentation.

→ We realized that this sentence is not clear. Thanks to suggest a better sentence! We revised it.

Lines 292-294 - "Although some strains showed good potential for wine-making through single co-fermentation with S. cerevisiae". I do not believe co-fermentations have been performed at this point in the manuscript?
"non-Saccharomyces yeasts are still considered better for producing more favorable wines and avoiding contamination of foreign microorganisms by reducing the fermentation time". Please detail some of the features referred to as "favorable wines". More importantly, as the authors state above, non-Saccharomyces yeasts typically take a long time to finish the fermentation process and produce small amounts of ethanol. Stating that these yeasts are considered better to avoid contamination and reduce fermentation time seems contradictory. If the authors mean that S. cerevisiae co-fermentations with non-Saccharomyces species contribute to enhance these properties vs only S. cerevisiae fermentations, they should reference studies supporting such claims.

→ We apologize that we missed checking these wrong sentences. During the proofreading process, these sentences were wrong edited. We corrected these sentences. Thanks for kindly pointing out them.

Figure 1 -  Units for the 0 to 5 scale in Figure 1 are required.

→ API ZYM kit represents the enzyme activity based on the intensity of color change so there is not a certain unit. We used a color reaction chart provided from a manufacturer and added additional explanation into material and method section 2.3. Thanks.

Section 3.5 - More discussion on the attained concentrations of produced metabolites is warranted. For example, the concentrations of isoamyl acetate or 2-phenylethyl acetate present in Table 4 are significantly higher than reported in other studies. Possible factors such as distinct fermentation conditions, medium composition, S. cerevisiae strains used, etc., that may justify the data should be discussed.

→ We think that the excessive amounts of some volatile ester compounds are derived from the wine starter, especially S. cerevisiae W-3 (W-3). Our previous studies that used W-3 also indicated that various wines contained high amounts of isoamyl acetate and 2-phenylethyl acetate. When we used other S. cerevisiae for winemaking, these compounds were very low as much as generally reported. You can find our previous studies by below links.

https://onlinelibrary.wiley.com/doi/full/10.1111/ajgw.12405

https://www.sciencedirect.com/science/article/pii/S0168160518307529

→ In addition, the present study is mainly focusing on non-Saccharomyces yeasts and their β-glucosidase activities for producing volatile terpene compounds in MBA wine. S. cerevisiae W-3 was used as a control strain. Therefore, the excessive volatile ester compounds are less important in this study because these compounds are not produced by non-Saccharomyces yeasts. Please kindly consider it, thanks.

Reviewer 2 Report

SB Lee and HD Park report W. 402 anomalus JK04 and M. pulcherrima S36 as potential starters for improving Korean MBA wine quality in the manuscript entitled “Isolation and investigation of potential non-Saccharomyces yeasts to improve the volatile terpene compounds in Korean Muscat Bailey A wine”.

The manuscript presents interesting data, in a very good scientific sound, however, some aspects need to be addressed:

  • The authors should explain why their analysis of the volatile terpenes in wines limits at only three of them.
  • The authors should present the results obtained for the concentration of these terpenes in a new column in Table 4 or another table as a percentage of the terpenes concentration in the control strain, in order to emphases the increased levels of volatile compounds.

Author Response

Dear Reviewer 2,

We appreciate your kind review for improving the manuscript.

We have revised the manuscript followed by your suggestions. All modifications in the manuscript indicate in red font with comments. Following are our detail responses against your comments.

SB Lee and HD Park report W. 402 anomalus JK04 and M. pulcherrima S36 as potential starters for improving Korean MBA wine quality in the manuscript entitled “Isolation and investigation of potential non-Saccharomyces yeasts to improve the volatile terpene compounds in Korean Muscat Bailey A wine”.

The manuscript presents interesting data, in a very good scientific sound, however, some aspects need to be addressed:

  • The authors should explain why their analysis of the volatile terpenes in wines limits at only three of them.

→ When we analyzed the volatile aromatic compounds using GC-MS, only three kinds of volatile terpene compounds were detected. We would guess a lot of factors such as grape cultivars, geological differences, soil, climates, water, nitrogen fertilization, or harvest times affect volatile aromatic compound formation including monoterpenes. We added this into results and discussion section (line 399-401). We hope this can meet your concern, thanks.

  • The authors should present the results obtained for the concentration of these terpenes in a new column in Table 4 or another table as a percentage of the terpenes concentration in the control strain, in order to emphases the increased levels of volatile compounds.

→ We added the percentages of three volatile terpene compounds of JK04 and S36 wines by comparing those of the control wine in Table 4. Thanks.

Reviewer 3 Report

Dear Authors,

I found the manuscript entitled ”Isolation and investigation of potential non-2 Saccharomyces yeasts to improve the volatile terpene 3 compounds in Korean Muscat Bailey A wine” an interesting work especially for the actors in the wine sector. 

Few observations I would like to point out:

In the experimental design, you specified the grapes were washed. That is fine for a lab-scale study, but what is your position with respect to the industrial processing of these wines? What are the limits because grapes are not washed when industrially processed. 

The study aim pointed especially to the impact on the volatile profile of wines, but the results regarding the sensory and volatile profiles were only briefly described. A multivariate analysis will be useful to better understand the impact of these yeast strains to wine volatile and sensory profiles. 

Conclusions must reflect the results, not general aspects. Please reformulate the conclusions based on this. Also try to include Table 4 in the Results and discussion section, not in the Conclusions.

Author Response

Dear Reviewer 3,

We appreciate your kind review for improving the manuscript.

We have tried to revise the manuscript as possible as we can. We hope the revised manuscript will be accepted by you. All the modifications in the manuscript indicated in red font with comments. Following are our detail responses against your comments.

Dear Authors,

I found the manuscript entitled ”Isolation and investigation of potential non-2 Saccharomyces yeasts to improve the volatile terpene 3 compounds in Korean Muscat Bailey A wine” an interesting work especially for the actors in the wine sector. 

Few observations I would like to point out:

In the experimental design, you specified the grapes were washed. That is fine for a lab-scale study, but what is your position with respect to the industrial processing of these wines? What are the limits because grapes are not washed when industrially processed. 

→ This is a great question. We are currently developing wine starters and transferring the strains to the local wineries. Industrially grapes are typically not washed because of the problem with a tremendous amount of grape. Therefore, acceptable amount of sulfur dioxide and activated wine starter are added to grape must to avoid bacterial contamination. We think that an excessive sulfur could inhibit the growth of non-Saccharomyces yeasts. Fortunately, W. anomalus JK04 and M. pulcherrima S36, two main strains in this study, showed high SO2 tolerances. Even though we haven’t yet determined the optimal conditions such as SO2 for an industrial trial, these two strains will be practicable for the wine industry without any contamination. Thanks.

The study aim pointed especially to the impact on the volatile profile of wines, but the results regarding the sensory and volatile profiles were only briefly described. A multivariate analysis will be useful to better understand the impact of these yeast strains to wine volatile and sensory profiles. 

→ We agree with the review’s comment. MANOVA (multivariate analysis of variance) can be a very useful tool to figure out the relationship between strains, wine aroma and sensory profiles. However, it is difficult to make a multivariate analysis and discuss about it for a short revision time. We found some references that can support our results more so we added more discussion about the relationship between wine starter and sensory profiles. Please be kind to understand and accept this. Thanks.    

Conclusions must reflect the results, not general aspects. Please reformulate the conclusions based on this. Also try to include Table 4 in the Results and discussion section, not in the Conclusions.

→ Thanks for giving us very good suggestion. The previous conclusion contained some general aspects so we removed them and added the actual results based on the present study, thanks.

Round 2

Reviewer 1 Report

The authors have addressed the issues raised by the reviewer in a satisfactory fashion and have provided appropriate rebuttals for all the comments. Most requested changes are now present in the manuscript and I have no further comments. 

Reviewer 3 Report

Dear Authors,

I agree with the publishing of the manuscript in its current content.